# Epoxidized and Maleinized Hemp Oil to Develop Fully Bio-Based Epoxy Resin Based on Anhydride Hardeners

**DOI:** 10.3390/polym15061404

**Published:** 2023-03-11

**Authors:** Alejandro Lerma-Canto, Maria D. Samper, Ivan Dominguez-Candela, Daniel Garcia-Garcia, Vicent Fombuena

**Affiliations:** 1Technological Institute of Materials (ITM), Universitat Politècnica de València (UPV), Plaza Ferrándiz y Carbonell 1, 03801 Alcoy, Spain; 2Instituto de Seguridad Industrial, Radiofísica y Medioambiental (ISIRYM), Universitat Politècnica de València (UPV), Plaza Ferrándiz y Carbonell 1, 03801 Alcoy, Spain

**Keywords:** epoxidized hemp seed oil, vegetable oils, crosslinking, cyclic anhydrides

## Abstract

The present work aims to develop thermosetting resins using epoxidized hemp oil (EHO) as a bio-based epoxy matrix and a mixture of methyl nadic anhydride (MNA) and maleinized hemp oil (MHO) in different ratios as hardeners. The results show that the mixture with only MNA as a hardener is characterized by high stiffness and brittleness. In addition, this material is characterized by a high curing time of around 170 min. On the other hand, as the MHO content in the resin increases, the mechanical strength properties decrease and the ductile properties increase. Therefore, it can be stated that the presence of MHO confers flexible properties to the mixtures. In this case, it was determined that the thermosetting resin with balanced properties and high bio-based content contains 25% MHO and 75% MNA. Specifically, this mixture obtained a 180% higher impact energy absorption and a 195% lower Young’s modulus than the sample with 100% MNA. Also, it has been observed that this mixture has significantly shorter times than the mixture containing 100% MNA (around 78 min), which is of great concern at an industrial level. Therefore, thermosetting resins with different mechanical and thermal properties can be obtained by varying the MHO and MNA content.

## 1. Introduction

Thermosetting polymers, particularly epoxy resins, are widely used in engineering as adhesives and polymeric matrix for composite materials. This product family currently accounts for 14% [1] of the global polymeric market, being used in industrial sectors such as the automotive, aircraft and marine industries, civil infrastructures, electronic components and sporting goods, among others [2,3]. Epoxy resins are generally characterized by excellent mechanical, thermal, adhesive and solvent resistance properties [4].

Aside from the properties above, epoxy resins have several drawbacks. On the one hand, most of them are obtained from petrochemical products, which, combined with the difficulty of being recycled, makes the reduction of the carbon footprint produced during their life cycle very difficult [5]. On the other hand, epoxy resin is characterized by its inherent brittleness due to its highly crosslinked structure and moisture sensitivity [6].

Acting on the first point, numerous studies have focused on the total or partial replacement of both main components, the conventional synthetic reinforcement and the resin, by natural fibers or bio-based polymers [7,8,9]. The fibers that are most commonly used to develop green composite are based on ramie, jute, sisal, kenaf, hemp or flax [8]. These studies cover a wide range of topics from manufacturing techniques, material properties, fiber pre-treatments, coatings or possible applications. On the other hand, petrochemical epoxy resins can be replaced by bio-based epoxy resins, formulated from precursors such as furans, tannins, cardanol, natural rubber and especially vegetable oils (VO) [10,11,12].

In this context, VO are of particular interest due to them being sustainable, inexpensive and easily substitutable with other petrochemical epoxy resins for specific low-load bearing applications, as well as the improvement of the fragility problems mentioned above [13]. VO are formed by a long chain structure based on unsaturated triglycerides (C=C), which can easily be modified by active molecules such as oxirane oxygen, maleic anhydride, hydroxyl or acrylate. These modified VO can be employed as plasticizers, chain extenders, compatibilizers, coatings and thermosetting resins [14,15,16,17]. Epoxidation is one of the most widely used processes in the chemical modification of VO, with commercially available epoxidized soybean (ESO) and linseed oils (ELO). These can be used as reactive diluents instead of styrene to produce vinyl esters or in applications such as coatings, automotive and steel primers, thermal insulation, glues and adhesives [18]. However, these oils are also used in food production, which may raise an ethical issue in the case of the growing demand for bio-based feedstock in engineering applications. Therefore, to alleviate this conflict of interest, it is essential to focus on using fast-growing non-food oil crops as feedstock for bioresins.

Hemp seeds contain between 28–35% oil, depending on the geographical region of cultivation, variety of seeds and climatic conditions [19]. The annual export value of hemp seed in Europe in 2021 amounted to 34,412 tonnes, with the Netherlands being the European country that exported the most hemp seed in 2021, with a total of 27,435 tonnes [20]. In addition, hemp seed oil, due to its high content of unsaturated fatty acids, allows for a wide variety of chemical modifications, such as epoxidation, maleinization or acrylation [21]. Therefore, it has been shown that hemp as a raw material at the industrial level is promising for the manufacture of bioresins. As it is possible to observe in Table 1, the theoretical oxirane oxygen content of the hemp oil, a parameter related to the amount of monounsaturated and polyunsaturated fatty acids (MUFA and PUFA, respectively), is above 10.53%, being one of the VO with the most significant potential. Currently, there are few studies in the literature reporting on the use of hemp oil in the manufacture of thermosets. Manthey et al. [22], due to the need to develop new bio-based materials, developed new biocomposites made from hemp oil epoxidized with jute fiber as a reinforcement, and then compared them with samples containing commercial epoxidized soybean oil. They showed that samples with epoxidized hemp oil had slightly better mechanical, water absorption and dynamic mechanical properties than samples that were manufactured with epoxidized hemp oil. Similarly, when mixing epoxidized hemp oil with jute fibers, the resulting material is a perfect competitor with commercially produced epoxidized soybean oil in biocomposite applications.

On the other hand, to convert epoxidized vegetable oil into a crosslinked thermoset material, hardeners such as anhydrides or amines are used [10,17]. Anhydrides are the most employed curing agents, but they can present problems due to a susceptibility to hydrolytic degradation [31]. This aspect can be directly related to the moisture sensitivity of the epoxy resin and, furthermore, it can be increased if the epoxy resin is used to develop composites with natural fibers, which can contain moisture contents between 3 and 13% [6].

Thus, to avoid this problem and look for environmentally friendly alternatives to these petrochemical hardeners, vegetable oils modified by a maleinization process can be employed. For this purpose, the unsaturations present in the triglyceride are reacted with maleic anhydride (MA) through a combination of Diels–Alder reactions and “ene” reactions. The maleic groups present in the maleinized vegetable oil can react with the epoxy groups of the epoxidized vegetable oil (EVO), resulting, with the right combination of accelerators and catalysts, in a crosslinked structure. Some previous studies have looked at maleinized linseed oil as a hardener [15] and maleinized castor oil to produce biopolymers [4]. Still, again, there is no literature related to the use of maleinized hemp oil. 

Therefore, the main aim of the present study is to develop and optimize an epoxy resin based on epoxidized hemp oil and use a different content of maleinized hemp oil as a potential substitute for the petroleum-derived methyl nadic anhydride (MNA). When using different proportions of maleinized hemp oil and MNA, a wide range of mechanical and thermal properties can be obtained. Furthermore, these different ratios will ultimately define the bio-based carbon fraction of the final material. Although the use of epoxidized hemp oils as a basis for future bio-based resins is still underexplored, the main novelty of this study is the use of maleinized hemp oil (MHO) as a bio-based hardener. The use of MHO opens the door to the substitution of current petrochemical-based hardeners, which are based on molecules such as anhydrides or acids, resulting in almost 100% bio-based thermoset resins.

## 2. Materials and Methods

### 2.1. Materials

The hemp seed was obtained from a local market in Callosa de Segura. A CZR-309 (Changyouxin Trading Co., Zhucheng, China) press machine at room temperature was used to extract the hemp seed oil (HSO).

The epoxidation process was carried out with acetic acid (99.7%), sulfuric acid (97%) and hydrogen peroxide (30% *v*/*v*) supplied by Sigma Aldrich (Sigma Aldrich, Madrid, Spain). On the other hand, the maleinization process was performed by adding maleic anhydride (MA) with purity >98% supplied by Sigma Aldrich (Madrid, Spain).

The hardeners used for crosslinking the epoxidized hemp oil were methyl nadic anhydride (MNA) and maleinized hemp oil (MHO). MNA, with an anhydride equivalent weight (AEW) of 178 g·eq^−^^1^, is of petrochemical origin and was supplied by Sigma Aldrich (Madrid, Spain). On the other hand, MHO is of biological origin. 

In addition, glycerol at 0.8 wt.% was used as an initiator and 1-methylimidazole at 2 wt.% was used as an accelerator, both of which were supplied by Sigma Aldrich (Madrid, Spain) [32]. Figure 1 displays the chemical structures of all the components used, such as the epoxy resin, the crosslinkers, the initiator and the accelerator.

### 2.2. Epoxidation Process

To carry out the epoxidation reaction to obtain the bio-based epoxy matrix, the process used by Dominguez-Candela et al. [23] was followed with minor modifications. In this case, a three-neck round-bottomed flask with a capacity of 1000 mL was used. Inside this flask, a two-bladed stirrer was placed and immersed in a thermostatic water bath. The temperature in the bath could be controlled at ±0.1 °C according to the desired temperature. The epoxidation process was carried out with a duration of 8 h at a temperature and constant agitation of 70 °C and 220 rpm, respectively. After 10 min, following arrival at the required temperature, a mixture of sulphuric acid and hydrogen peroxide was added dropwise. The addition of this mixture took 30 min to complete. Figure 2 shows the proposed reaction mechanism during the epoxidation stage of hemp oil. This mechanism will later be contrasted using different techniques such as oxiranic oxygen content or FTIR.

The oxiranic oxygen content of the resulting epoxidized hemp oil was 7.2, obtained according to ASTM D1652. On the other hand, the epoxy equivalent weight (EEW), which is defined as the mass (expressed in grams) of the epoxy resin containing one equivalent of the epoxy group (g·eq^−1^), of the EHO was obtained according to ASTM D1652 by titration and the value obtained was 226.

### 2.3. Maleinization Process

The process followed to obtain MHO from virgin hemp oil has been described in previous studies [16]. MHO is characterized by an acid number of 106 mg KOH g^−1^ and a maximum viscosity at 20 °C of 10 dPa·s.

### 2.4. Sample Preparation

The different formulations were made by keeping the amount of EHO, glycerol and 1-methyl imidazole constant and changing the amount of the hardeners, MNA and MHO, as seen in Table 2. The hardeners’ ratio of epoxide equivalent weight to anhydride equivalent weight (EEW:AEW) was set at 1:1 [32].

The different mixtures were placed in aluminum containers to be weighed, shaken vigorously and then poured into a silicone mold to obtain standard rectangular specimens (80 mm × 10 mm × 4 mm) according to ISO 178. The curing process of the samples was carried out at 90 °C for 3 h and post-curing at 120 °C for 1 h. A schematic representation of the preparation of the different samples can be seen in Figure 3.

Figure 4 shows the interaction of EHO when reacting with MHO. The free volume can be seen to increase due to the functionalized long fatty acid chains. Therefore, the chain mobility is increased, contributing to better flexibility of the final thermosetting resins [33]. As for the equivalences used for the EHO and MHO molecules, the dots shown in black refer to the oxirane groups in EHO and the red dots to the maleic groups in MHO.

Figure 5 shows the plausible reaction between EHO, used as an epoxy resin, and MNA, used as a crosslinker. In this case, the reaction is initiated by the interaction between a hydroxyl group present in the initiating agent molecules, with the MNA giving rise to an ester. The acid group resulting from this reaction reacts with an epoxy group to produce a diester and a new hydroxyl group [34]. The MNA confers rigidity to the final material, since, as can be seen in Figure 3, the resulting structure is more clustered than that seen in Figure 2.

### 2.5. Oxirane Oxygen Content (O_o_) and Acid Value

The oxirane content (*O_o_*) was determined according to ASTM D1652. For this purpose, the sample of EHO had to be dissolved in chlorobenzene, followed by a drop of crystal violet and titrated using a hydrobromic acid (HBr) solution in glacial acetic acid. The *O_o_* content was obtained using Equation (1):(1)$$O_{o}(\mathrm{wt}.\%)=1.6\times N_{i}\times \frac{\left(V-B\right)}{W}$$
where *N* is the normality of HBr to glacial acetic acid, *V* refers to the volume of HBr solution used in the titration of the sample (expressed in mL), *B* refers to the volume of the HBr solution used to perform the blank titration (expressed in mL) and *W* refers to the amount (in grams) of sample used. At least five measurements were made for the sample and the average values were reported.

The acid value was obtained according to ISO 660. A titration of hemp oil (2 g) dissolved in 5 mL ethanol was used to determine the acid value content using potassium hydroxide solution as an ethanolic solution standard reagent to a phenolphthalein end point (the pink color of henolphthalein persisted for at least 30 s). Finally, Equation (2) was used to obtain the acid value,
(2)$$Acid\;value=\frac{56.1\times V\times C}{m}$$
where *C* corresponds to the exact concentration of the potassium hydroxide (KOH) solution (expressed in mol·L^−^^1^), *V* is the KOH volume used for the sample titration (expressed in mL) and *m* refers to the mass of the sample used to carry out the titration (expressed in grams).

### 2.6. Fourier Transform Infrared Spectroscopy (FTIR)

Fourier transform infrared spectroscopy (FTIR) was used to analyze the chemical structure of the virgin hemp oil, EHO, MHO and MNA. These samples were analyzed using a Bruker Vector 22 spectrometer from Bruker Española, S. A. (Madrid, Spain). The analyzed samples were subjected to a total of 20 scans in the range of 4000–400 cm^−1^, with a resolution of 4 cm^−1^. The spectra obtained were normalized with a limit ordinate of 1 absorbance unit.

### 2.7. Mechanical Characterization

Flexure, impact and hardness tests were carried out to analyze the mechanical properties of the different solid mixtures. The flexural test was carried out with an Ibertest ELIB 30 universal testing machine from S.A.E. Ibertest (Madrid, Spain) at room temperature. A crosshead speed of 5 mm·min^−1^ was used.

The impact tests were performed using a 6 J Charpy pendulum from Metrotec S.A (San Sebastián, Spain) according to ISO 179-1.

The shore D hardness was obtained using a Shore D hardness durometer 676-D (J. Bot. S. A., Barcelona, Spain) according to ISO 868. Five samples were tested for each test to obtain an average.

### 2.8. Thermal Characterization

The curing process of the different samples was studied by differential scanning calorimetry (DSC). DSC tests were carried out in a Mettler Toledo 821e calorimeter (Schwerzenbach, Switzerland). Samples were subjected to a temperature ramp of 30–350 °C at a rate of 10 °C·min^−1^ under a nitrogen atmosphere, with a flow rate of 66 mL·min^−1^. The starting temperature, the maximum crosslinking temperature (peak) and the final temperature of the process were obtained from each of the calorimetric curves. In addition, the enthalpy value (ΔH) of each sample was obtained from the integration of the exothermic peak area.

On the other hand, DSC was carried out with the same equipment and conditions as described above but using the crosslinked samples to obtain the cure percentages of these. Given that the curing process of the samples was carried out at 90 °C for 3 h and 1 h at 120 °C, it is possible that the samples were not 100% cured. DSC curves allow for analysis to determine if a small exothermic peak appears, making it possible to measure the % cured by comparing the obtained enthalpies in the first curing cycle from 30 to 350 °C in the liquid samples, which are cured 100%, and the second, smaller exothermic peak.

### 2.9. Thermomechanical Characterization

Dynamic mechanical and thermal analysis was performed in the plate-plate mode in an oscillating rheometer AR G2 from TA Instruments (New Castle, DE, USA). The samples were in a liquid state and were subjected to an isothermal temperature of 90 °C for 5 h at a frequency of 1 Hz. The gel time was obtained as the crossover point between the storage modulus (G′) and the loss modulus (G″). In addition, the rheometer was also used in torsion mode to test the cured rectangular samples with dimensions of 40 mm × 10 mm × 4 mm. These samples were subjected to a temperature ramp from −20 to 110 °C at a frequency of 1 Hz, a heating rate of 2 °C·m^−1^ and a strain rate of 0.1%.

### 2.10. Morphological Characterization

After the impact test, the fractured samples were taken to observe their surfaces in a field emission scanning electron microscope (FESEM) model Zeiss ULTRA from Oxford Instruments (Abingdon, UK) with a voltage of 2 kV. Before observation, the samples were coated with a thin layer of gold and platinum using an EM MED020 sputter coater from Leica Microsystems (Wetzlar, Germany).

## 3. Results and Discussion

### 3.1. Chemical Properties

Figure 6 shows the FTIR spectra of virgin hemp oil, MHO, EHO and MNA. The characteristic peaks of the double bonds are number 1, located at 3010 cm^−1^ (=CH_(v)_), which is caused by the stretching of the cis-olefin bonds, number 2, which is located at 1672 cm^−1^ (C=C_(v)_) and is due to the stretching of disubstituted cis-olefins and number 3, located at 723 cm^−1^ (C=C _(cis-δ)_) and caused by the combination of out-of-plane deformation and oscillating vibration in cis-disubstituted olefins [35]. As can be seen in Figure 6b,c, corresponding to the MHO and EHO, respectively, these peaks are diminished compared to virgin hemp oil. This is due to the fact that these double bonds have reacted during the maleinization and epoxidation processes, thus reducing the number of double bonds present in them. On the other hand, it can be observed how in the MHO sample (Figure 6b), two peaks appear representing the virgin hemp oil sample, located at 1781 and 1861 cm^−1^ (peak 4) and related to the symmetrical and antisymmetric vibrations of the carbonyl (C=O) of the anhydride groups observed, respectively. This is due to the maleic anhydride used as a reagent in the maleinization process of virgin hemp oil [36]. This proposal has been supported by studies where maleinized chia oil, which is chemically very similar to MHO, has been analyzed using NMR [37]. In this study, it was concluded that the characteristic peak at 2.8–3.2 ppm is attributed to the methylene and succinic protons created after the maleinization of the oil. This confirms the presence of reactive maleic anhydride (MA) groups in the maleinized oil, which was used as a hardener for the first time. In the sample of EHO (Figure 6c), a new peak is observed at 821 cm^−^^1^ (peak 5) in comparison with virgin hemp oil, related to the existence of the oxirane group (COC _(v)_). This group appears due to oxygen insertion into the double bonds through the peracetic acid formed by epoxidation [38]. Finally, a peak can be seen appearing in all the samples at 3470 cm^−^^1^ (peak 6), which is associated with the elongation (v) of -OH. This peak is observed higher in the EHO sample (Figure 6c) because it is related to the vibration of the hydroxyl group and demonstrates the formation of -OH groups due to the opening of the epoxy ring in the epoxidation process [39]. 

### 3.2. Thermal Properties

Differential scanning calorimetry (DSC) was used to analyze the cure cycle of the different thermosetting resins studied. Figure 7 shows the DSC curves of the curing cycles of the different resins, where it is possible to identify the different exothermic peaks of the samples referred to in the crosslinking process. As can be seen, there are variations in the initial, peak and final temperatures of the reaction for the different samples. In this case, it can be observed that for the 100% MNA sample, the curing process takes place at higher temperatures than in the rest, with a starting temperature of 140 °C and a final process temperature of around 225 °C. However, when MHO is added to the samples, it is observed that the start and end temperature of the resin reaction decreases, with the decrease being higher as the MHO content in the sample increases. In this case, it is observed that the sample with a 100% MHO content presents a start and end of the cure temperatures of 111 °C and 210 °C, respectively, which means a decrease of 34 °C and 15 °C, representing the beginning and end of the cure temperatures of the 100MNA sample. As can be seen, this same trend can be observed in the peak temperature, referring to the maximum reaction rate. That is to say, it can be observed that when MHO is added to the sample, the maximum crosslinking temperature decreases, with this decrease being greater as the MHO content increases in the sample. However, it can be observed that the addition of 25% MHO (75MNA25MHO) hardly affects the maximum crosslinking temperature, obtaining the same peak temperature as the 100MNA sample. These results indicate that the EHO reaction with MHO takes place more easily than reactions using MNA, and these results are consistent with the gel time, as it has been observed that MNA leads to higher gel time values than MHO [40].

Similarly, the enthalpy shows the same downward trend, decreasing as the MHO content increases (Table 3). In this case, the maximum reaction enthalpy is obtained with the 100MNA mixture at a value of 189.40 J·g^−^^1^. On the other hand, the minimum enthalpy value of 83.33 J·g^−^^1^ is reached with the 100MHO sample. Therefore, MHO leads to a lower exothermicity. In addition, sample 75MNA25MHO also shows a decrease in enthalpy to values of 141.30 J·g^−^^1^. Less exothermic values are obtained due to the chemical structure of MHO, as these macromolecules increase in weight. Still, on the other hand, the number of epoxide groups per gram is lower than in MNA [41].

Regarding the percentage of curing of the samples, all the samples have a high percentage of curing. For the samples that contain MNA in their formulation, the curing percentage is at least 90%. On the other hand, the samples containing only MNO as a hardener have a curing percentage of 85%. This is a good sign as a result obtained from the sample with 100% MNA content does not differ significantly from the one with 100% MHO content. 

### 3.3. Thermomechanical Properties

As for the gel time, it is critical in handling thermoset materials as, at this point, the material stops flowing and cannot be processed. The rheometer was used to obtain this value, considering the gel time when a phase angle equal to 45° is obtained. Table 4 shows the most relevant data for this test: the start of the curing process (δ ≈ 90°), the end (δ ≈ 0°) and the intermediate point or gel time (δ = 45°). When looking at these results, it can be seen that as the amount of MHO increases, the reaction rate and crosslinking of these mixtures also increase, thus reducing the curing time. In this case, the sample containing 100% MNA (100MNA) has a crosslinking onset of 10,200 s, a gel time of 11,500 s and a crosslinking endset of 15,270 s. However, after incorporating 25% MHO, which refers to the sample 75MNA25MHO, it is observed that these values decrease, with a crosslinking start of 4690 s, a gel time of 5433 s and a crosslinking start of 7900 s. Finally, the sample containing 100% MHO as a hardener (100MHO) shows a crosslinking onset value of 555 s, a gel time of 1006 s and, finally, a crosslinking endset of 1920 s. This decrease in the curing time of the MHO-containing samples suggests that the anhydride group included in the MHO is more reactive than the MNA, so crosslinking is faster [42]. This reduction in the gelation time resulting from the presence of MHO is interesting in the field of composites as it reduces the precipitation of particles that could lead to phase separation and heterogeneous material [43]. Moreover, this reduction in curing times is an important feature at the industrial level as fully cured materials can be obtained with shorter curing times.

Once all the cured and post-cured EHO materials crosslinked with the MNA/MHO blends were characterized with the rheometer, it was observed that changing the amount of MNA/MHO in the samples resulted in a change in the glass transition temperature (T_g_) of the materials. The phase angle (δ) is shown in Figure 8 and the values obtained for the glass transition temperature (T_g_) are compiled in Table 5. As can be seen, T_g_ decreases as the amount of MHO in the MNA/MHO mixtures increases. The sample with the highest T_g_ is the 100MNA with a value of 48.7 °C. On the other hand, a significant decrease in T_g_ is observed with the addition of MHO, with the lowest T_g_ value being obtained for the sample with 100% MHO content at a value of 6.8 °C. Furthermore, this decrease in T_g_ after the incorporation of MHO results in materials with higher ductility. This is best observed in those samples with high MHO contents above 50%, where the T_g_ values are below room temperature. In conclusion, as this T_g_ decreases, it can be said that the material changes from being brittle to being a more ductile material. 

### 3.4. Mechanical Properties

The flexural strength and flexural modulus of the samples obtained are presented in Figure 9. As can be seen in Figure 9a, the flexural modulus of the 100MNA sample is around 300 MPa. On the other hand, samples containing MHO have a lower stiffness, which becomes lower as the percentage of MHO increases and the percentage of MNA decreases. Specifically, the presence of 75% MNA in the sample leads to a decrease in the flexural modulus, with a value of 100 MPa, representing a decrease of 195%. This decrease is greater as the MHO content in the sample increases, obtaining a modulus of 7 MPa for the sample with 100% MHO content.

Figure 9b represents the flexural strength of the tested samples. In this case, a similar trend to the flexural modulus is observed; that is, as the MHO content in the sample increases, the flexural strength decreases compared to the 100MNA sample. In this case, it can be seen how the flexural strength decreases from almost 7 MPa for the sample 100MNA to 1 MPa for the sample with the highest MHO content (100MHO). This decrease in mechanical strength properties is related to the chemical structure of the hardeners used since, on the one hand, MNA, which is a cyclic anhydride, confers rigidity to the mixture and, on the other hand, MHO, composed of triglycerides, provides flexibility to the samples containing it [37]. Rösch and Mülhaupt [44], obtained highly flexible and rubber-like crosslinking polymers using anhydrides (succinic, hexaphydrophthalic and norbornene dicarboxylic acid) as a bio-based epoxy matrix epoxidized soybean oil.

On the other hand, in Figure 10a, the results obtained for the Shore D hardness of the samples are shown. In this case, as in the case of flexural strength, it is observed that the hardness decreases as the MHO content increases in the samples. In this case, the maximum hardness is obtained for the crosslinked sample with 100% MNA, 63 Shore D. This value is very similar to that obtained in the bibliography when crosslinking soybean oil epoxidized with maleic anhydride, getting a value of 70 Shore D [45]. As the MHO content increases, this hardness decreases in the samples, with the lowest hardness value being obtained in the crosslinked sample with 100% MHO, with a value of 21 Shore D. Furthermore, it can be observed that the samples crosslinked with 75% MNA and 50% MNA obtain similar values of around 44 Shore D.

Finally, the results of energy adsorption in the different samples obtained after performing the Charpy impact test are shown in Figure 10b. As can be seen in Figure 10b, two of the five samples subjected to the impact test (25MNA75MHO and 100MHO) failed to break and, therefore, no values are available. In this case, it can be seen that the value obtained in the 100MNA sample, 6.3 kJ/m^2^, is lower than that obtained in the 75MNA25MHO sample, which is 17.6 kJ/m^2^, an increase of 180%. In addition, the 50MNA50MHO sample presents a very similar value to the previous sample, since the value obtained is 17 kJ/m^2^. This increase in impact energy absorption corroborates with the increase in ductility due to the presence of MHO in the samples. Sahoo et al. [46] reported that adding a mixture of an epoxy toughened with renewable resources, linseed oil and a bio-based crosslinker to a petroleum-based epoxy (DGEBE) increases the impact absorption energy by 40% over petroleum-based resin alone.

### 3.5. Morphological Properties

To support the results obtained in the mechanical tests, the fractured surfaces of the samples after the Charpy test were studied using field emission scanning electron microscopy (FESEM). Figure 11 shows the fracture FESEM images of the samples subjected to the Charpy impact test. The sample with 100% MNA content is shown in Figure 11a, and it can be seen that the fracture surface is smooth, which is characteristic of a rigid material. As can be seen in Figure 11b,c, when the MHO content in the samples increases, the surface is no longer smooth and cracks are observed on the fracture surface. This is directly related to the higher ductility that MHO confers to the samples. These results perfectly agree with those obtained in the mechanical tests, as they confirm the increased ductility as the MHO content increases. Similarly, Domínguez-Candela et al. [14] report that by increasing the oil content in the sample, the resulting mixture gives rise to a more ductile material.

## 4. Conclusions

After compiling all the data obtained, it can be concluded that maleinized hemp oil (MHO) is an excellent crosslinking agent next to methyl nadic anhydride (MNA), which is of petrochemical origin, for epoxidized hemp oil (EHO). After performing the mechanical tests, it was observed that the sample containing 100% MNA (100MNA) presented with high rigidity and brittleness, whereas, with the addition of MHO in different amounts, it was observed that the material showed greater ductility and flexibility. For example, a significant difference in mechanical properties was observed between the 100MNA sample and the sample with 25% MHO (75MNA25MHO), as it changed from a rigid material to a more ductile material with only the addition of 25% MHO. The decrease in the flexural strength of the sample with 100% MNA compared to the sample with 75% MNA and 25% MHO is 11%. On the other hand, the impact energy absorption for the same samples shows an increase of 180%.

On the other hand, after the calorimetric study, it was shown that the incorporation of MHO leads to a reduction in the curing temperatures of the different samples studied, obtaining lower starting, maximum crosslinking and end temperatures as the MHO content increases in the samples. For example, the 100MNA sample presents a start, maximum crosslinking and end temperature of 140 °C, 191 °C and 225 °C, respectively. On the other hand, if the other extreme is observed, which is the 100MHO sample, the values obtained are 111 °C, 155 °C and 210 °C for the start, maximum crosslinking and end temperature, respectively. The reduction in the maximum crosslinking temperature between these two samples is 23%. Furthermore, it has been observed that as the MHO content in the samples increases, the gel time decreases. It has been observed that the gel time is reduced from 11,500 s for the 100MNA sample to values of up to 1000 s for the 100MHO sample. In addition, with the results obtained for the T_g_ of the materials, it was observed that when the MHO content increased, the T_g_ decreased, which means that the material was changing from a more rigid to a more flexible one.

Once these results have been presented, it can be said that the mixture with the most balanced properties compared to the 100MNA sample is the 75MNA25MHO sample, as it has a higher renewable content, greater flexibility and a shorter curing time, and this is very interesting from an industrial point of view. The developed thermoset resins, due to their high ductile properties, could be used in a variety of industrial sectors, such as construction, automotive, aerospace, electronics and marine, among others. In construction, they can be used for the manufacture of insulation panels, coatings and adhesives. In the automotive and aerospace industries, they can be used in the production of structural parts and components for vehicular interiors. In electronics, they could be employed in the encapsulation and printed circuit board components and in the marine industry, in the manufacturing of saltwater-resistant parts. Finally, it should be said that different mechanical and thermal properties can be obtained by changing the percentage of MHO, as it is possible to get material with more or less flexibility and most importantly, with high renewable content, as this is an essential characteristic from an environmental perspective.

## Figures and Tables

**Figure 1 polymers-15-01404-f001:**
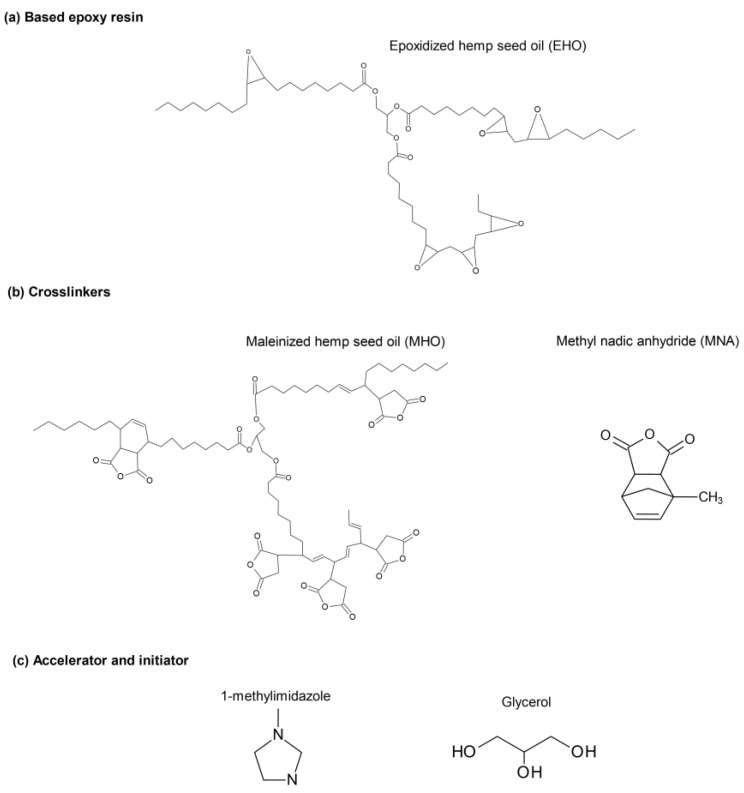
Chemical structures of the components used to obtain the final materials.

**Figure 2 polymers-15-01404-f002:**
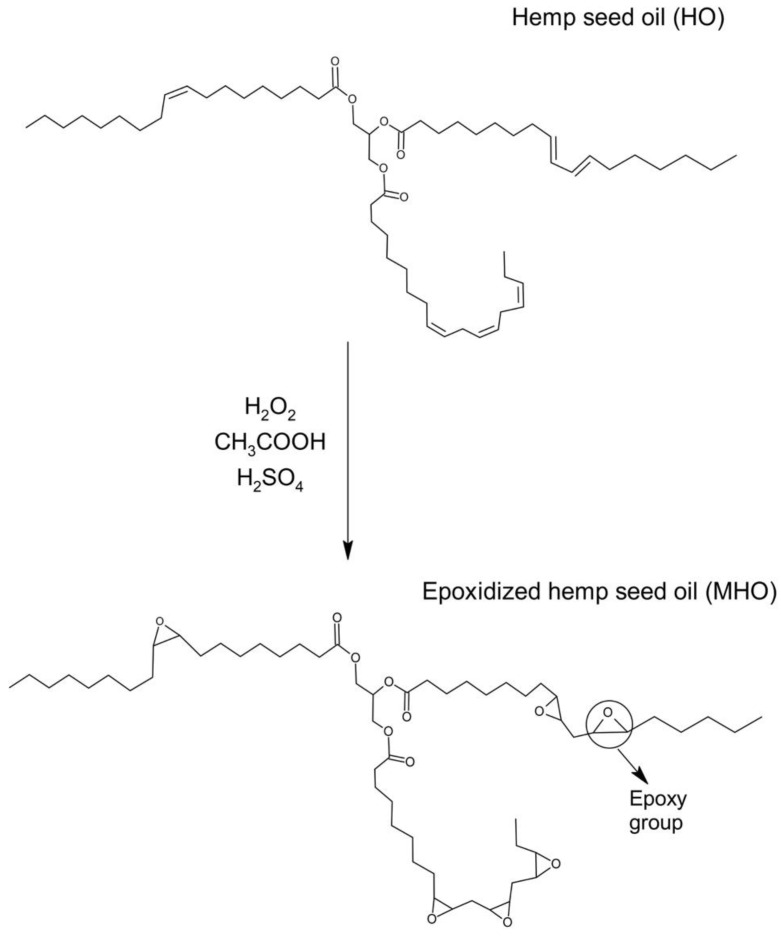
Schematic representation of the epoxidation reaction.

**Figure 3 polymers-15-01404-f003:**
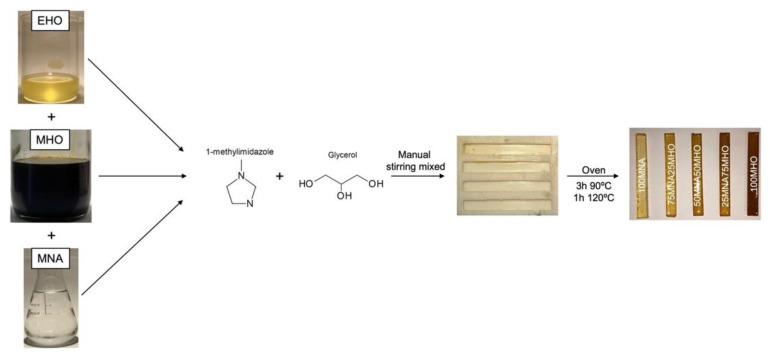
Diagram of sample preparation.

**Figure 4 polymers-15-01404-f004:**
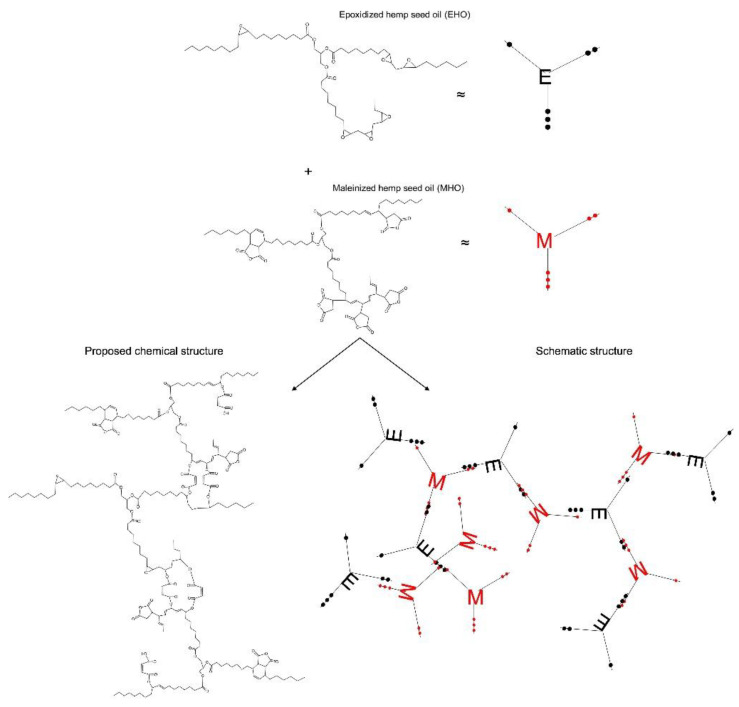
Epoxidized hemp oil (EHO) reaction using maleinized hemp oil (MHO) as a crosslinker. Black dots: oxirane groups; Red dots: maleic groups.

**Figure 5 polymers-15-01404-f005:**
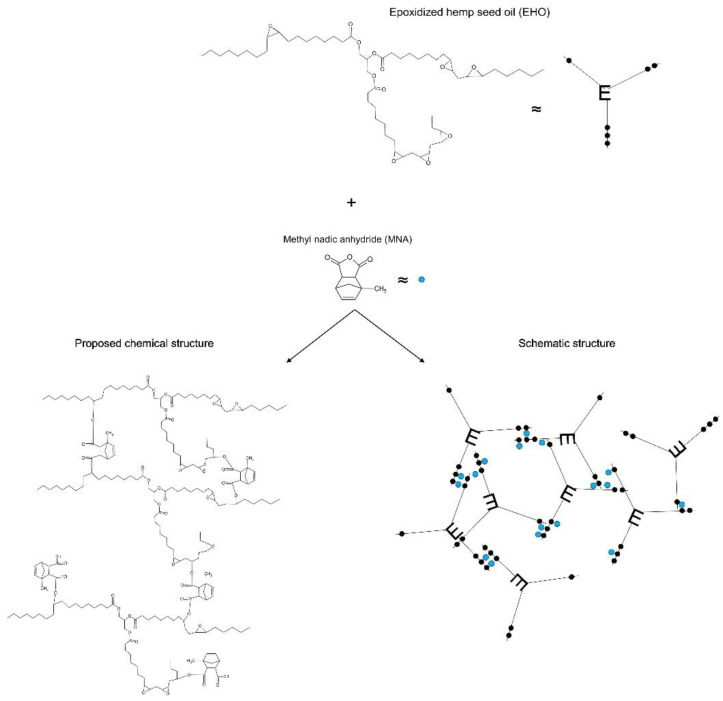
Epoxidized hemp oil (EHO) reaction using methyl nadic anhydride (MNA) as a crosslinker. Black dots: oxirane groups; Blue dots: MNA molecule.

**Figure 6 polymers-15-01404-f006:**
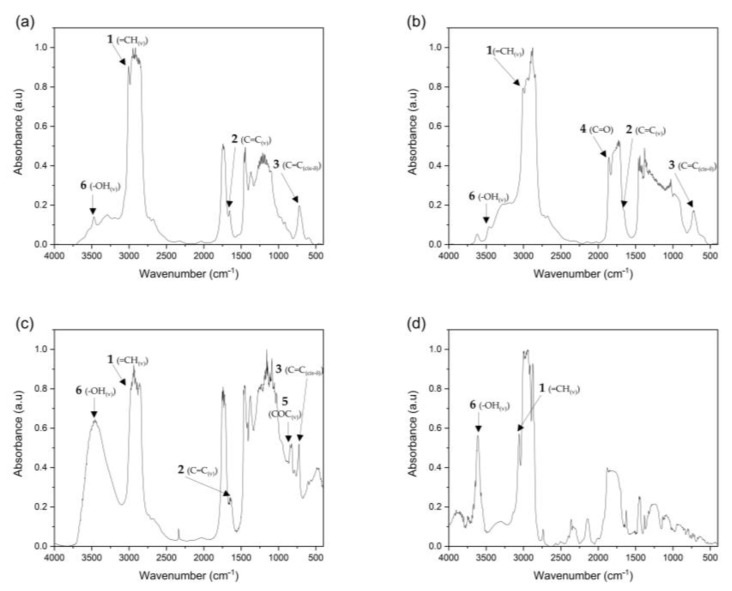
FTIR spectra of the different elements in the samples, (**a**) virgin hemp oil, (**b**) maleinized hemp oil (MHO), (**c**) epoxidized hemp oil (EHO) and (**d**) methyl nadic anhydride (MNA).

**Figure 7 polymers-15-01404-f007:**
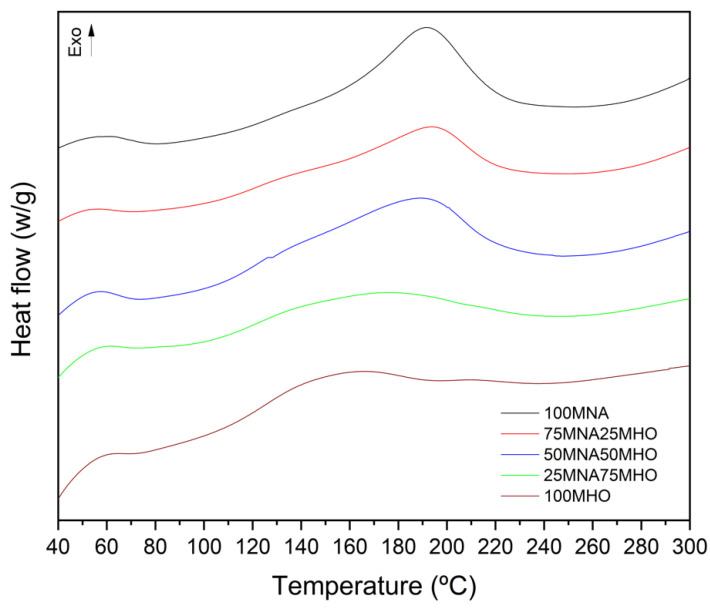
Comparison of the dynamic DSC thermograms of the curing process of different samples composed of the epoxidized hemp oil (EHO) and varying contents of methyl nadic anhydride (MNA) and maleinized hemp oil (MHO).

**Figure 8 polymers-15-01404-f008:**
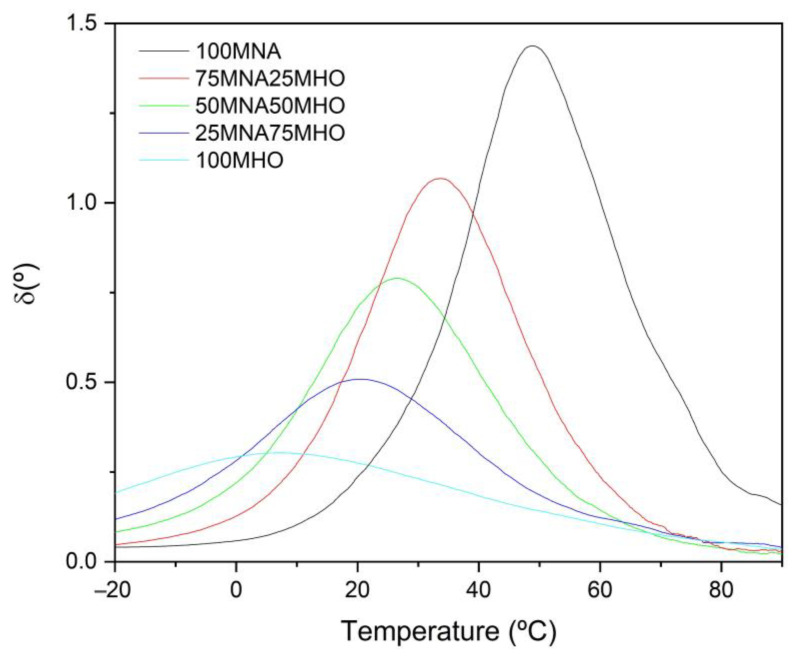
Phase angle variation for EHO-based resins crosslinked with a mixture of maleinized linseed oil (MHO) and methyl nadic anhydride (MNA).

**Figure 9 polymers-15-01404-f009:**
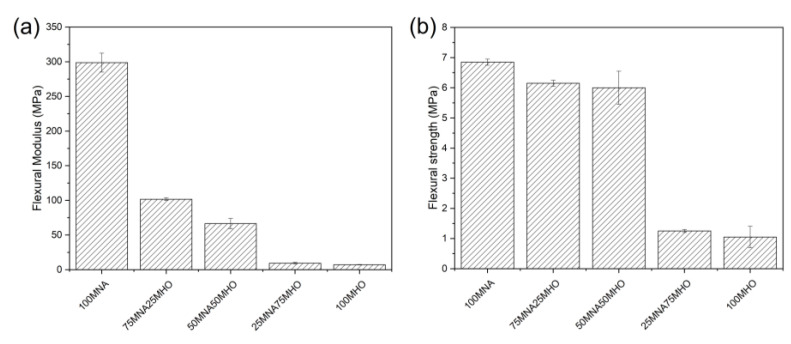
Results of flexural test on epoxidized hemp oil (MHO) crosslinking with different percentages of maleinized hemp oil and methyl nadic anhydride (MNA) (**a**) flexural modulus and (**b**) flexural strength.

**Figure 10 polymers-15-01404-f010:**
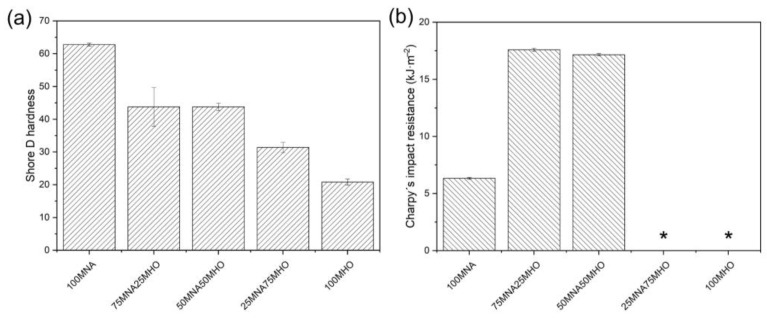
Results of impact and hardness tests on epoxidized hemp oil (MHO) crosslinking with different percentages of maleinized hemp oil and methyl nadic anhydride (MNA). (**a**) Shore D hardness and (**b**) Charpy’s resistance impact. * Unbroken samples.

**Figure 11 polymers-15-01404-f011:**
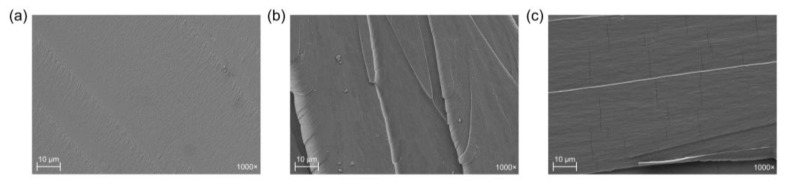
Field Emission Scanning Electron Microscopy (FESEM) images of the fracture surface of the samples at 1000×: (**a**) 100MNA, (**b**) 75MNA25MHO and (**c**) 50MNA50MHO.

**Table 1 polymers-15-01404-t001:** Oxirane oxygen content of different vegetable oils.

Vegetable Oils	Theoretical Oxirane Oxygen Content	References
Chia seed	11.05	[23]
Linseed	10.60	[24]
Hemp	10.53	Present study
Sunflower	7.57	[25]
Olive	7.41	[26]
Soybean	7.36	[27]
Corn	6.76	[28]
Castor	5.03	[29]
Palm	3.76	[30]

**Table 2 polymers-15-01404-t002:** Composition of different samples.

Code	MNA (%)	MHO (%)	Glycerol (wt.%)	1-Methylimidazole (wt.%)
100MNA	100	0	0.8	2
75MNA25MHO	75	25	0.8	2
50MNA50MHO	50	50	0.8	2
25MNA75MHO	25	75	0.8	2
100MHO	0	100	0.8	2

**Table 3 polymers-15-01404-t003:** Main parameters of the DSC test of the curing process of different samples composed of the epoxidized hemp oil (EHO) and varying contents of methyl nadic anhydride (MNA) and maleinized hemp oil (MHO).

Code	Onset (°C)	Peak Temperature (°C)	Endset (°C)	Enthalpy (J·g^−1^)	% Cure *
100MNA	145	191	225	189.4	93
75MNA25MHO	138	192	225	141.3	93
50MNA50MHO	122	188	216	135.4	90
25MNA75MHO	115	170	215	115.8	90
100MHO	111	155	210	83.3	85

* Data obtained in the determination of the percentage of cure of the crosslinked samples.

**Table 4 polymers-15-01404-t004:** Most relevant data obtained from the rheometer on the curing process of epoxidized hemp oil (EHO) crosslinked with methyl nadic anhydride (MNA) and maleinized linseed oil (MLO).

Code	Crosslinking Onset (δ ≈ 90°) (s)	Gel Time (δ = 45°) (s)	Crosslinking Endset (δ ≈ 0°) (s)
100MNA	10,200	11,500	15,270
75MNA25MHO	4690	5433	7900
50MNA50MHO	2000	2649	3690
25MNA75MHO	920	1350	2900
100MHO	555	1006	1920

**Table 5 polymers-15-01404-t005:** Values of glass transition temperature (T_g_) for EHO-based resins crosslinked with a mixture of maleinized linseed oil (MHO) and methyl nadic anhydride (MNA).

Code	T_g_ (°C)
100MNA	48.7
75MNA25MHO	33.7
50MNA50MHO	26.7
25MNA75MHO	20.1
100MHO	6.8

## Data Availability

Authors are total agree to share the research data published. In the case, this information will be required, it is possible to contact with the corresponding author, who kindly will reply all information required.

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
