# Peer review of "Epoxidized and Maleinized Hemp Oil to Develop Fully Bio-Based Epoxy Resin Based on Anhydride Hardeners"

_polymers, 2023, doi:10.3390/polym15061404_

Round 1
Reviewer 1 Report
This manuscript revolves around describing thermomechanical and Regio thermal properties of epoxy raisins. These polymers are very much useful in modern society and there has been a significant industrial push to develop them as well. In this regard the manuscript serves very well read and provide information to the respective community,
I have some minor comments which I recommend addressing before publication
1. In introduction, references should be more elaborative and clearly problem statement needs to be addressed. I would suggest citing work of S.S Jamari et al. for other seminal examples regarding seed oils
2. Chemical structures are not consistent throughout the manuscript. I would suggest authors take help from any commercial chem structure drawing software to make better quality diagrams and reaction mechanisms
3. Did the authors performed any NMR or mass spec to confirm the structure of crosslinks?
4. Mechanism is no very clear; authors should explain the mechanism of action of the reactions to have a broad audience range
5. Scale bar is missing in SEM images, please provide
Author Response
Comments to the Author:
“In introduction, references should be more elaborative and clearly problem statement needs to be addressed. I would suggest citing work of S.S Jamari et al. for other seminal examples regarding seed oils”
Response:
The authors agree with the reviewer's statement and therefore the authors have decided to insert the following sentence with the corresponding reference in the text.
“These can be used as reactive diluents instead of styrene to produce vinyl esters or in ap-plications for coatings, automotive and steel primers, thermal insulation, glues, and adhesives. [18].”
- Islam, M.R.; Beg, M.D.H.; Jamari, S.S. Development of vegetable‐oil‐based polymers. Journal of applied polymer science 2014, 131.
“Chemical structures are not consistent throughout the manuscript. I would suggest authors take help from any commercial chem structure drawing software to make better quality diagrams and reaction mechanisms”
Response:
In accordance with the recommendations described by the reviewer, the authors have changed Figures 4 and 5. Now it is possible to observe a clearer reaction mechanism. For this, the authors have added the proposed chemical structure during cross-linking, when using MNA or MHO as the hardener. Since triglyceride-based molecules are large (approximately 950 g/mol), the authors maintain the explanation of the chemical structure through the schematic procedure, where it is possible to observe a greater amount of linked structures.
“Did the authors performed any NMR or mass spec to confirm the structure of crosslinks?”
Response:
The NMR technique is complicated to apply to already crosslinked samples, mainly due to the lack of solubility of the thermoset structure created after crosslinking. This makes it impossible to analyze the structures created by NMR. However, taking into account the reviewer's proposal, the authors have added the following reference:
“Ivan Dominguez-Candela, Aina Perez-Nakai, Elena Torres-Roca, Jaime Lora-Garcia and Vicent Fombuena. Development of a novel epoxy resin based on epoxidized chia oil as matrix and maleinized chia oil as bio-renewable crosslinker. J Appl Polym Sci. 2023;140:e53574”.
In this reference, a maleinized chia oil, very similar to MHO, is studied by NMR. In this study, it is concluded that the characteristic peak at 2.8-3.2 ppm is attributed to the methylene and succinic protons created after the maleinization of the oil. This confirms the presence of reactive maleic anhydride (MA) groups in the maleinized used for the first time as hardener.
This information has been added to the main manuscript and the reference cited.
Moreover, due to these limitations, the authors have analyzed the chemical interaction using other techniques such as FTIR. The obtained results have been presented in Figure 6, where new information regarding the reactive groups has been added.
“Mechanism is no very clear; authors should explain the mechanism of action of the reactions to have a broad audience range”
Response:
The authors thank the reviewer for the suggestion. In order to reach the largest possible audience, the authors have added information on the reaction mechanism in Figures 4 and 5. As previously mentioned, the possible interaction mechanism and the influence of using MHO or MNA as the hardener have been added.
To support this mechanism, the following information has been added.
Explanation regarding the use of MHO as hardener (Figure 4):
“Figure 4 shows the interaction of EHO when reacting with MHO. The free volume can be seen to increase due to the functionalized long fatty acid chains. Therefore, the chain mobility is increased, contributing to better flexibility of the final thermosetting res-ins [33]. As for the equivalences used for the EHO and MHO molecules, the dots shown in black refer to the oxirane groups in EHO and the red dots to the maleic groups for MHO.”
Explanation regarding the use of MNA as hardener (Figure 5):
“Figure 5 shows the plausible reaction between EHO, used as an epoxy resin, and MNA, used as a crosslinker. In this case, the reaction is initiated by the interaction between a hydroxyl group, present in the initiating agent molecules, with the MNA giving rise to an ester. The acid group resulting from this reaction reacts with an epoxy group to produce a diester and a new hydroxyl group [34]. The MNA confers rigidity to the final material, since, as can be seen in Figure 3, the resulting structure is more clustered than that seen in Figure 2.”
“Scale bar is missing in SEM images, please provide”
Response:
According to the reviewer, the scale bar in Figure 9 has been added and in addition, the magnification to which the SEM was performed has also been added (1000x).
“Figure 11. Field Emission Scanning Electron Microscopy (FESEM) images of the fracture surface of the samples at 1000x: a) 100MNA, b)75MNA25MHO and c) 50MNA50MHO.”

Reviewer 2 Report
I have the following suggestions for further improvement of the paper.
Comment 1. Give the schematic of the reaction in Epoxidation Process
comment 2. Mark the peaks in the FTIR spectra
comment 3. 2.9. Thermomechanical characterization, it is not DMTA, it is a rheometer. one cannot follow a cure reaction using DMTA. Please correct.
Comment 4. Please do tensile properties.
Comment 5. Please show the photographs of the sample preparation.
Comment 6. Please draw a schematic showing the reasons behind changing the properties of the sample.
Comment 7. Please write a novelty statement in the introduction.
Comment8. State some possible applications of the composites at the conclusion part.
comment 9. what is the % cure of the samples. The authors may do a dynamic DSC and find out the extend of crosslinking in the systems.
Author Response
Comments to the Author:
“Give the schematic of the reaction in the Epoxidation Process”
Response:
Following the reviewer's instructions, the authors have added a new figure (Figure 2) that details the reaction mechanism of the hemp oil epoxidation process.
Furthermore, the following information has been added to the manuscript.
“Figure 2 shows the proposed reaction mechanism during the epoxidation stage of hemp oil. This mechanism will be later contrasted using different techniques such as oxiranic oxygen content or FTIR.”
“Mark the peaks in the FTIR spectra.”
Response:
According to the reviewer´s comment, the authors have modified the FTIR spectra showed in Figure 6. Now it is possible to find in the Figure the information about the chemical bounds causing the characteristic peaks of the spectra. On the other hand, this information has been contrasted with other techniques of characterization, as for example, NMR. In this context, next sentence has been added:
“This proposal has been supported by studies where maleinized chia oil, chemically very similar to MHO, was analyzed using NMR. In this study, it was concluded that the char-acteristic peak at 2.8-3.2 ppm is attributed to the methylene and succinic protons created after the maleinization of the oil. This confirms the presence of reactive maleic anhydride (MA) groups in the maleinized used for the first time as hardener.”
“Thermomechanical characterization, it is not DMTA, it is a rheometer. one cannot follow a cure reaction using DMTA. Please correct.”
Response:
Following the suggestion by the reviewer, DMTA has been changed by rheometer.
“Please do tensile properties.”
Response:
The authors have not conducted tensile tests on the samples for the following reasons:
Thermoset resins, once cured, have a crosslinked structure that makes them rigid and brittle, making it difficult for them to undergo plastic deformation and resist tensile stress. In contrast, thermoplastic resins, lacking this crosslinked structure, are more flexible and resistant to tensile stress, making them more suitable for such tests. Therefore, if the aim of the study is to evaluate the mechanical strength of materials based on thermoset resins, the authors consider tests such as flexural, Charpy impact, or hardness to be more appropriate. Multiple studies based on thermosetting resins rely on mechanical characterization through flexural testing.
“Please show the photographs of the sample preparation.”
Response:
In accordance with the reviewer's suggestion, the authors have introduced a new Figure (Figure 3), where a schematic representation of the preparation of the different samples has been added. In this figure it is possible to see the different chemical products mixed, the mold employed, the conditions of the curing process, as well as, an image of the final appearance of the specimens.
The following information accompanies the interpretation of the image.
“The different mixtures were placed in aluminum containers to be weighed, shaken vigorously and then poured into a silicone mold to obtain standard rectangular speci-mens (80 × 10 × 4 mm3) according to ISO 178. The curing process of the samples was car-ried out at 90 °C for 3h and post-curing at 120 °C for 1h. A schematic representation of the preparation of the different samples can be seen in the Figure 3.”
“Please draw a schematic showing the reasons behind changing the properties of the sample.”
Response:
By the reviewer's proposal, the authors have modified the Figures 4 and 5. The reaction between epoxidized hemp oil (EHO) and hardeners has been showed in these Figures. The chemical structure proposed during the crosslinking process. Now it is possible to observe a clearer reaction mechanism. For this, the authors have added the proposed chemical structure during cross-linking, when using MNA or MHO as the hardener. Since triglyceride-based molecules are large (approximately 950 g/mol), the authors maintain the explanation of the chemical structure through the schematic procedure, where it is possible to observe a greater amount of linked structures.
To support this mechanism, the following information has been added.
Explanation regarding the use of MHO as hardener (Figure 4):
“Figure 4 shows the interaction of EHO when reacting with MHO. The free volume can be seen to increase due to the functionalized long fatty acid chains. Therefore, the chain mobility is increased, contributing to better flexibility of the final thermosetting res-ins [33]. As for the equivalences used for the EHO and MHO molecules, the dots shown in black refer to the oxirane groups in EHO and the red dots to the maleic groups for MHO.”
Explanation regarding the use of MNA as hardener (Figure 5):
“Figure 5 shows the plausible reaction between EHO, used as an epoxy resin, and MNA, used as a crosslinker. In this case, the reaction is initiated by the interaction between a hydroxyl group, present in the initiating agent molecules, with the MNA giving rise to an ester. The acid group resulting from this reaction reacts with an epoxy group to produce a diester and a new hydroxyl group [34]. The MNA confers rigidity to the final material, since, as can be seen in Figure 3, the resulting structure is more clustered than that seen in Figure 2.”
“Please write a novelty statement in the introduction.”
Response:
According to the reviewer´s comment, the authors have decided to make a short statement about the article's novelty in the introduction.In order to clarify this information, the following sentence has been added in the manuscript.
“Although the use of epoxidized hemp oils as a basis for future biobased resins is still un-derexplored, the main novelty of this study is the use of maleinized hemp oil (MHO) as a biobased hardener. The use of MHO opens the door to the substitution of current petro-chemical-based hardeners, which are based on molecules such as anhydrides or acids, resulting in almost 100% biobased thermoset resins..”
“State some possible applications of the composites at the conclusion part.”
Response:
The reviewers have added possible applications of the composites obtained in the conclusion section. The following sentence have been added to the manuscript:
“The developed thermoset resins, due to their high ductile properties, could be used in a variety of industrial sectors, such as construction, automotive, aerospace, electronics, marine, among others. In construction, they can be used for the manufacture of insulation panels, coatings, and adhesives. In automotive and aerospace, they can be used in the production of structural parts and components for the vehicle interior. In electronics, could be employed in the encapsulation and printed circuit board components and in the marine industry, in the manufacturing of saltwater-resistant parts.”
“what is the % cure of the samples. The authors may do a dynamic DSC and find out the extend of crosslinking in the systems.”
Response:
Following the reviewer's comments, the authors have studied the % cure of the samples through DSC.
The information of this new assay has been added in the section 2.8. Thermal characterization:
- In section 2.8. Thermal characterization:
“On the other hand, DSC was carried out with the same equipment and conditions as described above, but using the crosslinked samples to obtain the percentage of cures of these. Given that the curing process of the samples is carried out at 90ºC for 3 hours and 1 hour at 120ºC, it is possible that the samples are not 100% cured. DSC curves allow for analysis to determine if a small exothermic peak appears, making it possible to measure the % cured by comparing the obtained enthalpies in the first curing cycle from 30 to 350ºC in the liquid samples, which are cured 100%, and the second smaller exothermic peak.”
The interpretation of the results obtained are summarized in Section 3.2. Thermal properties:
“Regarding the percentage of curing of the samples, all the samples have a high percentage of curing. The samples that contain MNA in their formulation, the curing percentage is at least 90%. On the other hand, the sample containing only MNO as a hardener has a curing percentage of 85%. This is a good sign as a result obtained from the sample with 100% MNA content does not differ significantly from the one with 100% MHO content.”

Round 2
Reviewer 2 Report
The authors have carefully revised the paper based on my comments. The paper may be accepted in Polymers